# Understanding electrochemical switchability of perovskite-type exsolution catalysts

Alexander K. Opitz [1✉], Andreas Nenning[1], Vedran Vonk [2], Sergey Volkov[2], Florian Bertram[2], Harald Summerer[1,3], Sabine Schwarz[4], Andreas Steiger-Thirsfeld[4], Johannes Bernardi [4], Andreas Stierle [2] & Jürgen Fleig [1]

Exsolution of metal nanoparticles from perovskite-type oxides is a very promising approach to obtain catalysts with superior properties. One particularly interesting property of exsolution catalysts is the possibility of electrochemical switching between different activity states. In this work, synchrotron-based in-situ X-ray diffraction experiments on electrochemically polarized $La_{0.6}Sr_{0.4}FeO_{3-\delta}$ thin film electrodes are performed, in order to simultaneously obtain insights into the phase composition and the catalytic activity of the electrode surface. This shows that reversible electrochemical switching between a high and low activity state is accompanied by a phase change of exsolved particles between metallic $\alpha$-Fe and Fe-oxides. Reintegration of iron into the perovskite lattice is thus not required for obtaining a switchable catalyst, making this process especially interesting for intermediate temperature applications. These measurements also reveal how metallic particles on $La_{0.6}Sr_{0.4}FeO_{3-\delta}$ electrodes affect the $H_2$ oxidation and $H_2O$ splitting mechanism and why the particle size plays a minor role.

[1] TU Wien, Institute of Chemical Technologies and Analytics, Getreidemarkt 9/164-EC, 1060 Vienna, Austria. [2] Deutsches Elektronen-Synchrotron (DESY), 22607 Hamburg, Germany. [3] TU Wien, Institute of Materials Chemistry, Getreidemarkt 9/165-PC, 1060 Vienna, Austria. [4] TU Wien, University Service Centre for Transmission Electron Microscopy (USTEM), Wiedner Hauptstraße 8-10, 1040 Vienna, Austria. ✉email: alexander.opitz@tuwien.ac.at

For changing the world's energy supply towards sustainable resources, storage capabilities are required to buffer the irregular occurrence of renewable energy. Storing energy chemically as hydrocarbons is a very attractive option, since it could make use of large parts of the existing distribution infrastructure[1,2]. The first step of such energy storage processes is most likely electrolysis of water or carbon dioxide, which can be done in solid oxide cells with highest thermodynamic efficiency. The water splitting kinetics of these cells can be significantly enhanced by employing a novel type of oxide-supported catalysts, commonly denoted as exsolution catalysts[3–12]. This type of catalyst is based on catalytically active metal particles being prepared by reduction of perovskite-type oxides, which contain reducible transition metals. Upon reduction, the transition metal is exsolved from the perovskite lattice and forms metallic precipitates on the oxide surface[3,4,7,8,13–20]. In contrast to deposition of a catalytically active metal on top of an oxide, this approach yields nanoparticles that are socketed in the oxide and thus much less prone to particle growth at elevated temperatures[3,21]. Moreover, some studies even claim that such catalysts offer reversibility by redissolving the catalytically active material into the host-oxide upon an oxidative heat treatment[17,19,20,22,23].

Applying exsolution catalysts as electrodes in solid oxide electrolysis cells leads to a further degree of freedom, since here the exsolution of the catalytically active metal can also be triggered by the applied voltage[7,13–16,24]. Such an electrochemical polarization causes a lowering of the oxygen chemical potential inside the oxide electrode and thus a reduction of the cathode material leading to metal exsolution. Accordingly, the catalytic activity of the exsolution catalyst can be switched electrochemically[7,25]. Additionally, exsolution catalysts also offer promising opportunities for heterogeneous catalysis and may also be employed in the further steps of storing renewable energy— e.g., in power to fuel reactions. There, the possibility of electrochemically switching the activity or selectivity of the catalysts is extremely attractive, since it may offer fundamentally novel possibilities in fuel synthesis by simply adjusting the applied voltage.

For electrochemical water splitting on perovskite-type electrodes the appearance of metallic particles on the oxide surface was correlated with a step change in electro-catalytic activity[7,11,25,26]. On $La_{0.6}Sr_{0.4}FeO_{3-\delta}$ (LSF) electrodes, simultaneously performed electrochemical experiments and near ambient pressure X-ray photoelectron spectroscopy (NAP-XPS) revealed that water electrolysis rates strongly increase upon formation of $Fe^0$ particles and turn back to lower values as soon as the exsolved iron is reoxidized[26]. However, in this and other studies it remained unclear whether or not reoxidation also causes reintegration of the exsolved metal into the oxide, and the exact reason for the reversible switching effect is thus not understood yet.

The main goal of the present work is clarifying how exsolution catalysts can be reversibly switched between a high and low activity state upon changing the applied electrochemical voltage. To answer this question, synchrotron-based, surface sensitive X-ray diffraction (XRD) was employed on electrochemically polarised LSF thin film model electrodes. High temperature $H_2O$ electrolysis/$H_2$ oxidation was chosen as the model reaction probing the electrode's catalytic activity. The in-situ XRD measurements allowed a direct correlation of exsolved phases to the electro-catalytic activity of the LSF electrodes, thus unravelling the processes behind the switching effect. Moreover, our results also allow a mechanistic discussion of $H_2$ oxidation/$H_2O$ splitting on LSF surfaces with and without exsolved metallic particles and therefore contribute to an in-depth understanding of perovskite-type electrodes in general.

## Results and discussion

**In-situ X-ray diffraction on polarised LSF electrodes.** Electrochemical experiments at $625 \pm 10\,°C$ and simultaneous surface sensitive XRD measurements were performed at the beamline P07 of synchrotron DESY (Hamburg, Germany) with the setup sketched in Fig. 1 in an atmosphere containing 2.5% $H_2$, 0.25% $H_2O$, balance Ar (total pressure 1 bar). The electrochemical cell consisted of an yttria stabilized zirconia (YSZ) single crystal as electrolyte, an LSF thin film working electrode (WE) on top and a porous Pt/LSF counter electrode (CE) on the bottom side. In-situ surface XRD was performed on the ca. 250 nm thin LSF WE, deposited on a buried Ti/Pt current collecting thin film grid (for preparation of electrodes see Methods). A photon energy of 79.5 keV and a shallow incident angle of $\mu = 0.06°$ were used to facilitate a high surface sensitivity of the diffracted signal.

Typical 2D diffraction patterns are shown in Fig. 2. To remove adventitious carbon and to record a reference on the pristine samples, the very first heating of the model-type electrochemical cells was done in an oxidizing atmosphere with 5% $O_2$ (balance Ar). The corresponding diffraction pattern is depicted in Fig. 2a. The main information is then gained from the diffraction patterns obtained from polarised LSF films in $H_2$/$H_2O$; an example is given in Fig. 2b. For a detailed phase analysis please refer to Supplementary Fig. 1.

**Electrochemical switching between two activity regimes.** To monitor the catalytic activity of the LSF thin film electrodes, water electrolysis/hydrogen oxidation was chosen as a model reaction. From several previous experiments we know that the electrochemical reactions in $H_2$/$H_2O$ atmosphere are kinetically limited by a reaction step at the LSF surface[27–29]. Upon applying a voltage to the cell, a net electrochemical reaction takes place at the WE according to Eq. (1), which is written here in Kröger–Vink

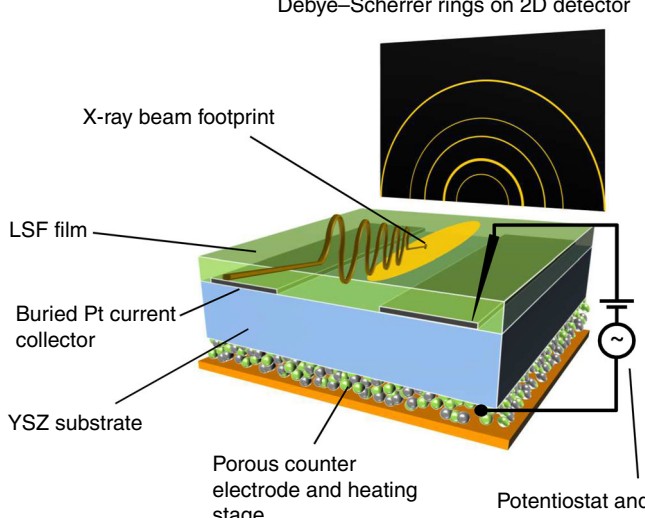

Debye–Scherrer rings on 2D detector

X-ray beam footprint

LSF film

Buried Pt current collector

YSZ substrate

Porous counter electrode and heating stage

Potentiostat and impedance analyzer

**Fig. 1 Sketch of the experimental setup for simultaneous electrochemical and X-ray diffraction measurements at elevated temperatures.** The sample—consisting of Y-stabilised zirconia electrolyte (blue), buried Pt current collector (gray), $La_{0.6}Sr_{0.4}FeO_{3-\delta}$ thin film working electrode (green), and porous Pt/$La_{0.6}Sr_{0.4}FeO_{3-\delta}$ counter electrode (green and gray particles at the bottom side)—is placed onto the heating stage (orange) within the experimental chamber (not shown for the sake of simplicity). The X-ray beam (symbolised by the brownish wave) hits the working electrode with a very flat angle of incidence. The diffracted X-rays are recorded as Debye–Scherrer rings on a 2D detector (outside the chamber), which is placed in a certain distance to the sample.

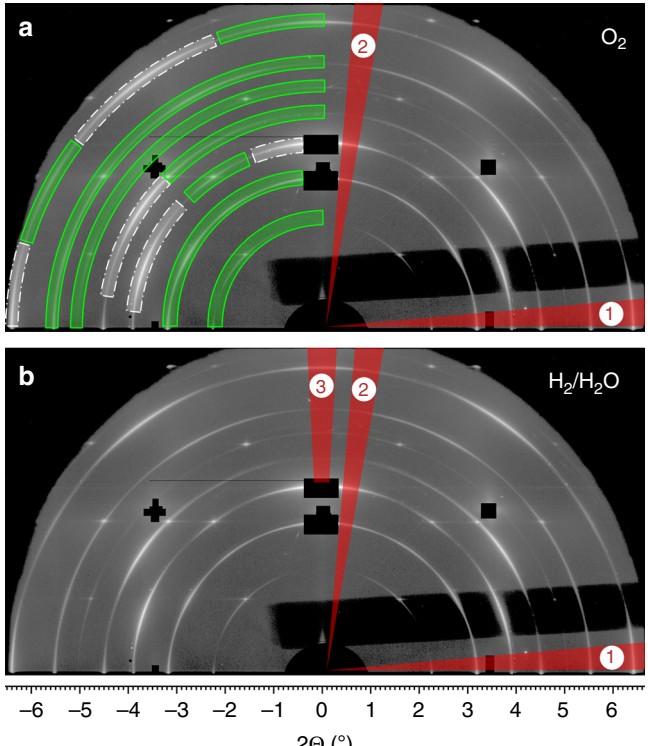

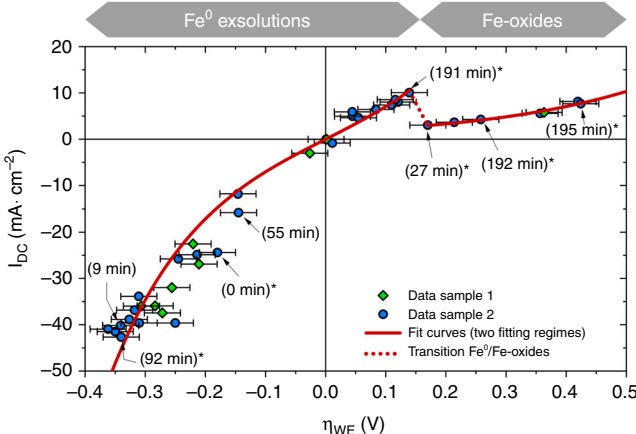

**Fig. 3 Plot of the area-related current versus the overpotential dropping at the working electrode.** The different symbols denote measurements on two different samples. Error bars (confidence intervals) for the voltage are comparatively large owing to a non-negligible polarisation resistance of the counter electrode; error bars for the current are smaller than the symbols. The solid red curves show the fit results of the high and low activity regime (details see supporting info). The dotted line is a guide to the eye showing the switching between both regimes. Since the set voltage was changed randomly between different values (including sign changes), the italic numbers in brackets specify the time after first observation of exsolution—for those with an asterisk diffraction data are shown in Figs. 4 and 5. The bars above the plot indicate the observed particles decorating the La$_{0.6}$Sr$_{0.4}$FeO$_{3-\delta}$ surface at the respective overpotentials.

**Fig. 2 2D diffraction patterns measured on pristine La$_{0.6}$Sr$_{0.4}$FeO$_{3-\delta}$ thin film electrodes with buried Pt current collectors at 625 ± 10 °C in two different atmospheres.** The black parts are beam stops (either physical or digital ones) to block signal with a too high intensity. **a** Atmosphere: 5% O$_2$/balance Ar. The circular segments in the left part of the pattern indicate different phases (gray/dash-dotted: Pt, green/solid: La$_{0.6}$Sr$_{0.4}$FeO$_{3-\delta}$), while the numbered red triangles show different integration regions. **b** Atmosphere: 2.5% H$_2$/0.25% H$_2$O/balance Ar. For the sake of better visibility the phase identification was omitted here. On patterns measured under reducing conditions a third integration region was applied for comparison with samples after Fe exsolution.

notation:

$$H_2 + O_O^\times + 2Fe_{Fe}^\times \rightleftharpoons H_2O + v_O^{\cdot\cdot} + 2Fe_{Fe}^{/}. \qquad (1)$$

Therein, $O_O^\times$, $v_O^{\cdot\cdot}$, $Fe_{Fe}^\times$, and $Fe_{Fe}^{/}$ denote regular lattice oxygen (i.e., oxide ions), oxygen vacancies, regular lattice iron (i.e., Fe$^{3+}$), and reduced iron as a polaron (i.e., Fe$^{2+}$), respectively. In contrast to liquid-phase electrochemistry, the redox reaction at the LSF surface is not driven by an electrostatic potential difference[28,30]. Rather, the overpotential at the WE ($\eta_{WE}$) causes a defect chemical polarization of the LSF phase. This induces a change of the oxygen chemical potential and thus of the effective oxygen partial pressure in the bulk of the WE. Consequently, the defect chemistry of the LSF electrode is not in equilibrium with the gas phase, which causes a net rate of the reaction in Eq. (1). This net reaction rate is directly proportional to the measured electric current, in accordance with Faraday's law. The overpotential at the LSF electrode is thus the driving force for the reaction in Eq. (1)—the higher the resulting reaction rate (i.e., the current) is for a given $\eta_{WE}$, the higher is the electro-catalytic activity of the WE.

In Fig. 3 the area-normalized current ($I_{DC}$) is plotted versus $\eta_{WE}$ for two different samples, which are indicated by different symbols. For calculation of $\eta_{WE}$ from applied set voltages please refer to the Methods section. Each current value is an average over at least 20 s of steady-state current (usually even more) for a certain applied voltage. The resulting I–V-curve reflects all overpotential/current pairs measured during the synchrotron-

based in-situ XRD experiment. Since the entire duration of the experiment was rather long and since perovskite surfaces are known to be prone to degradation effects, the curve was not measured conventionally by step-wise increasing the applied voltage. Rather, the voltage was changed in a random pattern and even the sign of polarity was often changed back and forth. Thus, we could exclude the existence of a severe time dependence e.g., due to degradation effects. In additional lab-based experiments essentially the same I–V-curves could be measured with much more data points (see Supplementary Fig. 2).

In the cathodic regime the resulting curve shows a nonlinear behavior, which seems to continue to moderate anodic overpotentials. However, at $\eta_{WE}$ larger than about +150 mV the current drops sharply and also exhibits a less pronounced nonlinearity for higher anodic overpotentials. Such a decrease of the current for increasing driving force is very unusual and cannot be explained by standard (solid state) electrochemical models. Rather, we face two regimes with very different electrochemical activity and it is readily possible to jump from the high activity to the low activity curve and back—which we refer to as reversible electrochemical switching. A more detailed analysis of the I–V-curve is given in the Methods section, which also reveals that the step in the current–voltage curve is associated with a change of the reaction mechanism at the LSF surface. Moreover, impedance spectra were recorded on virgin as well as exsolved LSF electrodes, which are consistent with the DC data, thus also verifying the two activity regimes (see Supplementary Fig. 3).

**Activity regimes and chemical nature of exsolved particles.** In order to correlate the two activity regimes with the evolution of surface phases, the 2D diffraction patterns in Fig. 2 were analysed for different applied overpotentials. Some selected 2D patterns are shown in Supplementary Fig. 4 and the corresponding 1D patterns (obtained by integration of 2D data over χ) are depicted in

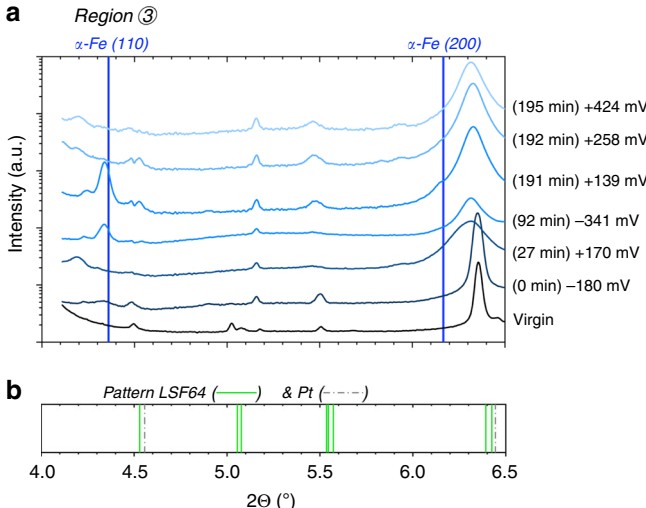

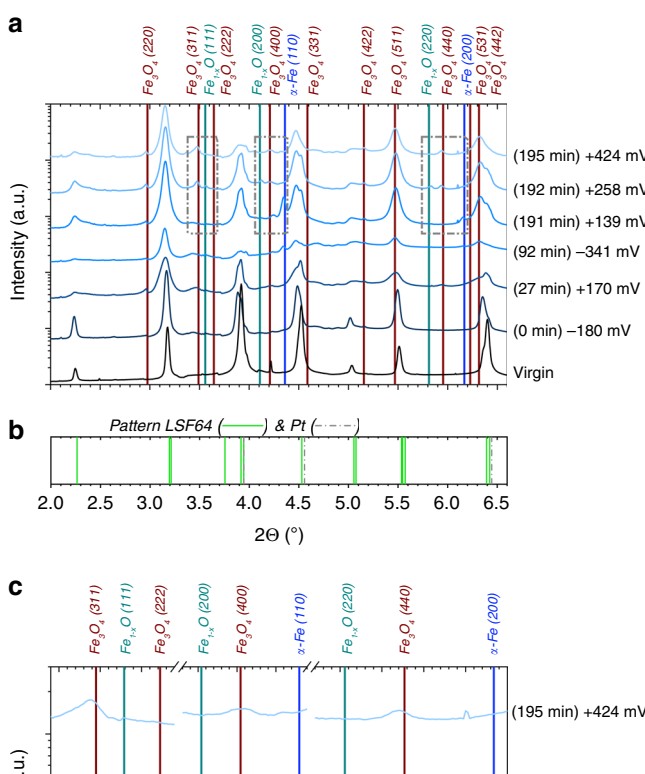

**Fig. 4 1D diffraction pattern obtained by integration within region 3 of the 2D patterns in Fig. 2b and Supplementary Fig. 4. a** The shown 2Θ part focuses on the region of possible reflexes of metallic iron. The blue vertical lines indicate the two expected reflexes of bcc iron (α-Fe, pdf# 04-014-0360)[31], which is the thermodynamically stable Fe modification at the experimental temperature. Please note that the scale of the intensity axis is logarithmic. **b** Literature pattern of rhombohedral $La_{0.6}Sr_{0.4}FeO_{3-\delta}$. (pdf# 00-049-0285) and platinum (pdf# 00-004-0802)[31].

Figs. 4 and 5. These patterns correspond to different points in the I–V-plot in Fig. 3 marked by the times in brackets (which indicate the chronological order of the measurements). These diffraction patterns clearly show significant differences in the phase composition of the LSF surface, seen by the appearance of additional reflexes—i.e., Debye–Scherrer rings (in 2D) or peaks (in 1D).

Figure 4a shows the 1D pattern obtained by integration within region 3 (cf. Fig. 2b and Supplementary Fig. 4) together with the expected positions of the Bragg reflexes of α-Fe (bcc) in this 2Θ range[31]. Since the very first exsolution of iron from the virgin LSF film only occurred after applying a sufficiently high voltage (details see below), characterization of the undecorated LSF electrode was possible even under conditions where metallic iron is thermodynamically stable (bottom pattern in Fig. 4a). Appearance of an α-Fe diffraction peak was triggered by applying a cathodic polarization of 180 ± 30 mV (0 min in Figs. 3, 4, and 5). A more accurate determination of the initial exsolution point was performed in lab-based electrochemical experiments (see section Methods and Supplementary Fig. 5). After its initial appearance, the signal from metallic iron showed two changes: either it increased from measurement to measurement or—for significant anodic polarizations—it completely vanished.

First, we consider the increase of the $Fe^0$ signal with time: See the growing Fe (110)-peak in Fig. 4a for the measurements from 0 min to 191 min, with an interim vanishing at 27 min (upon stronger anodic polarisation). This behavior already suggests that iron is not (completely) reintegrated into the perovskite lattice upon oxidation, i.e., oxidation does not correspond to a restart of the experiment. Moreover, Fe exsolution causes some changes of the LSF lattice (visible by broadening and shifts of some LSF-related reflexes), and those also persist under strong anodic polarization. Here, we focus on the continuous growth of the metallic particles by analysing the evolution of the peak width using Scherrer's formula[32] (Eq. (12), see Methods section). In Supplementary Fig. 6a the obtained particle size d is plotted versus the time from first appearance of exsolved $Fe^0$ particles. As soon as metallic particles exist on the surface—i.e., after applying

**Fig. 5 1D diffraction pattern obtained by integrating the entire 2D pattern in Fig. 2 and Supplementary Fig. 4. a** The vertical lines indicate the expected reflexes of bcc iron (α-Fe, pdf# 04-014-0360), $Fe_{1-x}O$ (pdf# 01-079-2175), and $Fe_3O_4$ (pdf# 01-080-6410)[31]. The dashed boxes indicate zoom regions. Please note that the scale of the intensity axis is logarithmic. **b** Literature pattern of rhombohedral $La_{0.6}Sr_{0.4}FeO_{3-\delta}$. (pdf# 00-049-0285) and platinum (pdf# 00-004-0802)[31]. **c** Zoom of the dashed boxes from (**a**).

a sufficiently high cathodic overpotential ($\eta_{WE} \approx 180$ mV, 0 min) —an increase in particle size with time is found, even under slightly anodic polarization (e.g., point at 191 min).

To visualise the surface particles formed by iron exsolution, scanning and transmission electron microscopy (SEM and TEM) images were recorded. These investigations were performed on a sample in the post-exsolved state subsequent to synchrotron studies and additional lab-based electrochemical characterization; after the initial exsolution the sample was exposed to reducing conditions for ca. 60 h. Figure 6a depicts an SEM image of the LSF thin film WE close to a current collector stripe. It exhibits a polycrystalline structure, as observed earlier on these films[27]. Since the image was recorded in secondary electron mode, the bright spots can be identified as protrusions that stick out from the surface. In the TEM image in Fig. 6b these protrusions can be identified as particles decorating the surface of the film. One of these particles is shown in the high resolution image in Fig. 6c. In the center of the particle and the neighboring LSF film, the average distance of the optically most prominent lattice planes was measured as 0.206 and 0.280 nm, respectively (see

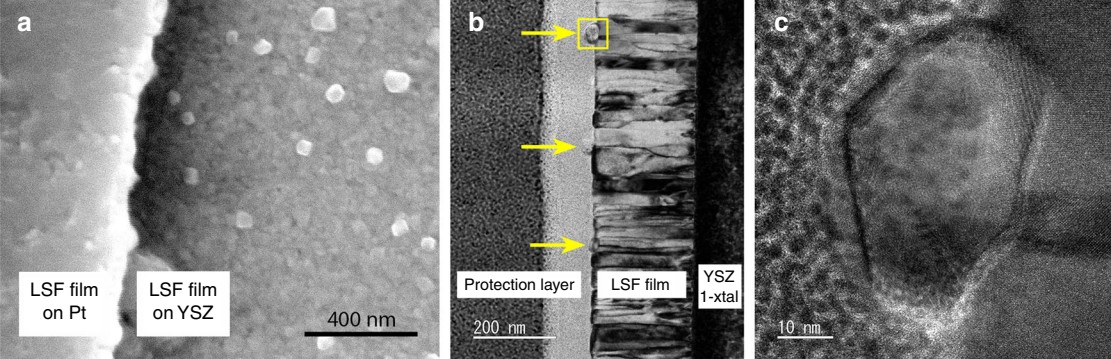

**Fig. 6 Electron microscopic images of La$_{0.6}$Sr$_{0.4}$FeO$_{3-\delta}$ thin films ca. 60 h after initial exsolution. a** Scanning electron microscopy image (secondary electron detector). The bright spots are the exsolved particles with an average diameter of 54 nm (standard deviation 16 nm). **b** Transmission electron microscopy bright field image of the film cross section. The marbled appearance of the columnar film is due to diffraction contrast. Exsolved particles are marked by arrows, the particle in the box is magnified in (**c**), which shows a high resolution image of this region. The slightly blurred looking halo-like fringe around the particle is due to magnetic interaction of the particle with the electron beam.

Supplementary Fig. 7), which is in good agreement with literature values for the α-Fe (110) lattice plane and the (110)/(104) planes of rhombohedral LSF, respectively[31]. This is a further evidence supporting the interpretation that the exsolved particles indeed consist of metallic bcc iron.

The average particle size after 60 h under reducing conditions was obtained from the SEM image in Fig. 6a to be 54 nm. This value is also included in Supplementary Fig. 6a and the entire data set of particle growth—in-situ and ex-situ—was analysed by employing kinetic models described in literature (for details please refer to the Methods section)[33]. As a result, we suggest a reaction-controlled growth with a significant effect of the strain of the particles, in line with studies focussing on the kinetics of the exsolution process[21,34]. Interestingly, this increase of the exsolution particle size is not reflected in the current–voltage curve, since differences of reaction rates (i.e., currents) for different times (e.g., at 9 and 92 min in Fig. 3) are within the error of the measured data. This allows for mechanistic conclusions, see below.

Next, we focus on the vanishing of the signal from metallic iron in Fig. 4. As mentioned above, the metallic iron is still present for low anodic overpotentials (e.g., +139 mV; 191 min), but applying a sufficiently high anodic overpotential (e.g., +170 mV, 27 min or +258 mV, 192 min) causes the α-Fe reflexes to disappear. The behavior of exsolved iron under such oxidizing conditions is revealed by the diffraction patterns depicted in Fig. 5. Under sufficiently high anodic polarization (+258 mV, 192 min) the exsolved iron is oxidized to Fe$_{1-x}$O (wustite) and Fe$_3$O$_4$ (hematite) and at even higher anodic overpotential (+424 mV, 195 min) it exists only as Fe$_3$O$_4$. This result thus clearly shows that for the experimental temperature of 625 °C large parts of iron exsolved as metallic particles under sufficiently reducing conditions do not reintegrate into the perovskite lattice under oxidising conditions.

**Analysis of the switching behavior.** Assuming the very first iron oxidation to proceed via the reaction Fe + ½ O$_2$ ⇌ FeO we can calculate the oxygen partial pressure of bulk Fe$^0$ oxidation and according to Nernst's equation (see Methods section and Eq. (7) therein) this corresponds to an anodic overpotential of +56 ± 5 mV for the transition between Fe$^0$ and FeO[35]. The experimentally obtained value for this transition point is defined by the step in the I–V-curve at +150 ± 46 mV (the relatively large error arises from the breadth of the transition region and the error of overpotential determination). For the significant deviation

between the expected and experimentally obtained value it might be particularly important that even though the WE is electrochemically polarised to a more oxidizing state, the gas phase itself is still at a chemical potential favouring metallic iron. The switching point at a higher anodic polarization than expected is thus seen as an indication of a kinetic competition between the oxidizing driving force from electrochemically polarised LSF and the reducing driving force of the gas atmosphere. This interpretation is also supported by electrochemical I–V-curves recorded in our lab (see Fig. 7 as well as Supplementary Fig. 2). From these measurements a switching point of +100 ± 10 mV was obtained, again significantly above the thermodynamic value of the Fe/FeO transition. Moreover, the switching behavior depends on the sequence of applied bias steps. For a Cycles voltage program a hysteresis is found (Fig. 7a), while this is not the case for a Butterfly measurement (Fig. 7b). This further supports our assumption of a kinetic interplay between chemical reaction of the particles with the gas phase and electrochemically triggered switching between Fe$^0$ and Fe oxide.

From both the in-situ and ex-situ experiments we can thus conclude, that while the removal of iron from the perovskite lattice is largely irreversible at the experimental temperature of 625 °C (and below), the transition between metallic and oxidized iron particles is completely reversible. This result proves that an electrochemically switchable catalyst can also be realized with the exsolved transition metal remaining on the perovskite surface. This finding is especially interesting, since at the typical temperatures of heterogeneously catalyzed reactions—such as (reverse) water gas shift, methane formation, and others—a redissolution of the catalytically active transition metal in the perovskite lattice is extremely slow. As shown here, a switching functionality can though be achieved at moderate temperatures by changing the oxidation state of the surface decorating particles. The optimum application temperature of switchable exsolution catalysts is thus expected to be in a moderate temperature regime. There, the growth of the particles is expected to be very slow, while the conductivities of LSF and an appropriate electrolyte (such as YSZ) are sufficient to allow for an electrochemical switching of the exsolved surface particles.

**Mechanistic discussion of the electrode reaction.** From all results shown above we conclude that the high activity regime is caused by the Fe$^0$ particles decorating the LSF surface. In the low activity regime these particles persist but as iron oxide instead. Both situations are indicated in Fig. 3 by the gray bars above the

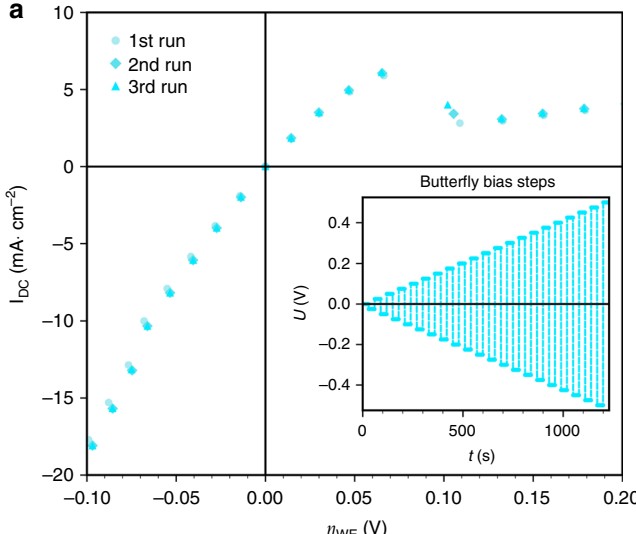

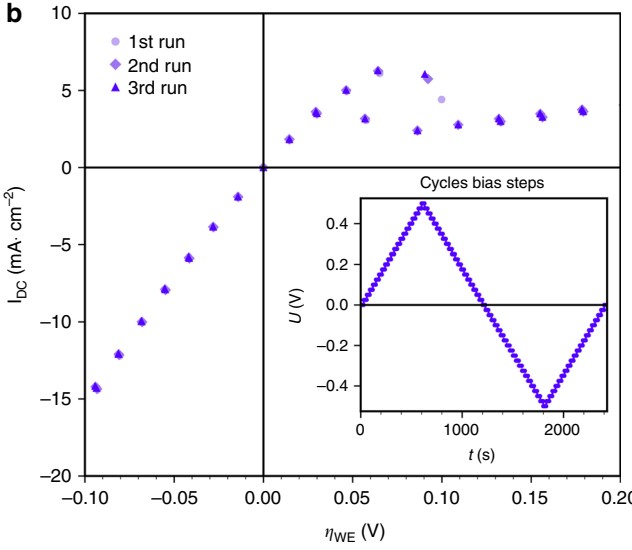

**Fig. 7 Current density versus overpotential curves measured in the lab on a post-exsolved sample—i.e., after in-situ experiments.** Error bars (confidence intervals) of overpotential and current are smaller than the symbols. **a** Result with voltage program Butterfly. **b** Result with voltage program Cycles. Both voltage programs are sketched as inset in the respective diagram.

I–V-plot. Switching from the low activity regime to high activity is thus not mainly caused by the exsolution of new $Fe^0$ particles. Rather, it proceeds primarily via reduction of the already existing iron oxide particles on the electrode surface. Even thought subsequent growth of the $Fe^0$ particles takes place, the size of the particles plays only a minor role for the electrochemical activity enhancement. Hence, the presence of some metallic iron seems sufficient to switch the LSF electrodes into the high activity state, although there are most likely significant changes in triple-phase boundary length associated with $Fe^0$ particle growth. This is a strong indication for a significant change in the reaction mechanism, which is associated with the sheer presence of metallic iron, see further discussion below.

The electrochemical promotion effect of $Fe^0$ particles exsolved from LSF was also found in a previous XPS study[26], but it remained unclear whether or not iron is reintegrated into the perovskite lattice under anodic (oxidizing) conditions. The

present results clearly show that exsolved iron is not (completely) reintegrated into LSF and, hence, we can conclude that indeed the $Fe^0$ particles are responsible for the enhanced catalytic activity rather than an iron-depleted perovskite.

In order to further interpret the atomistic reasons behind the observed switching of the electro-catalytic activity, we need to have a look on the mechanism of hydrogen oxidation/water splitting (Eq. (1)) on pure LSF. On oxides, hydrogen often adsorbs by an oxidative dissociation process forming surface hydroxyls[36,37]. Also on perovskite-type oxides chemisorption of hydrogen usually proceeds together with an electron transfer as shown in DFT studies on $SrTiO_3$[38] and $LaFeO_3$[39,40]. Thus we suggest oxidative hydrogen dissociation as the first step of $H_2$ reduction also on LSF:

$$H_2 + O_O^\times + Fe_{Fe}^\times \rightleftharpoons (OH)_O^\cdot + H_{ad,LSF} + Fe_{Fe}^/. \tag{2}$$

The remaining atomic hydrogen is oxidized in the next step, forming another surface hydroxyl:

$$H_{ad,LSF} + O_O^\times + Fe_{Fe}^\times \rightleftharpoons (OH)_O^\cdot + Fe_{Fe}^/. \tag{3}$$

Finally, two surface hydroxyls recombine, releasing water:

$$2(OH)_O^\cdot \rightleftharpoons O_O^\times + v_O^{\cdot\cdot} + H_2O. \tag{4}$$

Since the dissociative adsorption and desorption of water on a perovskite-type oxide was shown to be very fast in ref. [41], the entire reaction in Eq. (4), or any elementary step of it, is excluded as rate determining step of the sketched reaction scheme. Consequently, one of the reactions in Eqs. (2) or (3) represents the rate limiting step of the hydrogen oxidation/water splitting reaction on a perovskite-type electrode such as LSF.

On metallic iron, hydrogen usually adsorbs dissociatively

$$H_2 \rightleftharpoons 2H_{ad,Fe} \tag{5}$$

and this step is rather fast[37,42–46]. The atomic $H_{ad,Fe}$ very likely exchanges with $H_{ad}$ on LSF by surface diffusion. This process is commonly referred to as a spillover:[47,48]

$$H_{ad,Fe} \rightleftharpoons H_{ad,LSF}. \tag{6}$$

Please note that such a spillover process does not affect the equilibrium coverage of the $H_{ad}$ species on the LSF surface, which is defined by the chemical potential of hydrogen and cannot be affected by the presence of $Fe^0$ particles. However, upon an electrochemical polarization causing a net reaction, a hydrogen spillover from $Fe^0$-particles allows sustaining a sufficiently high $H_{ad,LSF}$ coverage on LSF, which leads to a higher net reaction rate (i.e., a higher DC current) via Eq. (3) than in case of undecorated LSF, where electrochemical polarisation causes a significant depletion of the $H_{ad,LSF}$ species.

As a consequence of the spillover, a bypass of the oxidative dissociation of $H_2$ (Eq. (2)) is established by $Fe^0$ (see sketch in Fig. 8). Actually, the step in Eq. (2) is the only elementary step that can be bypassed by a $H_{ad}$ supply via the metallic $Fe^0$ particles, since any other reaction step on the LSF surface is connected in series with Eq. 2. Moreover, the fact that metallic iron particles enhance the electrode reaction rate also rules out an alternative rate limiting step on LSF with $H_2$ being oxidized to two surface protons simultaneously $(H_2 + 2O_O^\times + 2Fe_{Fe}^\times \rightleftharpoons 2(OH)_O^\cdot + 2Fe_{Fe}^/)$. This could not be circumvented by a spillover species from the metallic particles and thus cannot lead to the behavior observed in Fig. 3. Hence, dissociative $H_2$ oxidation (Eq. (2)) has to be the rate limiting step on the bare LSF electrode. Since Fe-oxides do not facilitate dissociative hydrogen adsorption, this bypass is eliminated as soon as the exsolved iron particles are oxidized. On LSF decorated with metallic $Fe^0$ particles, the oxidation of $H_{ad}$ on LSF is most probably the rate limiting step (Eq. (3)). Here, the low

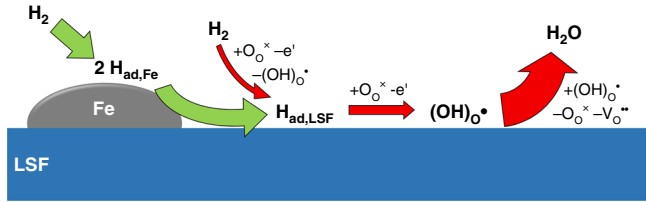

**Fig. 8 Sketch of the proposed mechanism for the case of H₂ oxidation. Reaction steps with red arrows proceed on the La₀.₆Sr₀.₄FeO₃₋δ surface, those with green arrows require the metallic iron particles.** The thicknesses of the arrows are a qualitative indication for the respective reaction rate.

surface concentration of $Fe_{Fe}^{\times}$ (i.e., $Fe^{3+}$) in LSF under reducing conditions[29] may play a crucial role. This combination of a $H_{ad}$ spillover from $Fe^0$ to LSF and a rate determining Eq. (3) would also explain the irrelevance of the $Fe^0$ particle size discussed above. As soon as a certain $Fe^0$ surface area is available and the density of $Fe^0$ particles is within the decay length of the spillover process, the $Fe^0$ particle size does not play a major role.

## Methods

**Sample preparation.** First, porous CEs were prepared on the unpolished bottom side of (100) oriented YSZ single crystals (9.5 mol% $Y_2O_3$, Crystec, GER) by consecutively brushing $La_{0.6}Sr_{0.4}FeO_{3\delta}$ (Sigma-Aldrich) and Pt (Gwent Electronics) paste, which were sintered together at 850 °C. For preparation of LSF thin film WE with buried current collector, 10 nm Ti and 100 nm Pt (99.95% pure) were deposited by magnetron sputtering (MED020 Coating system, BAL-TEC) on the polished top side of the YSZ crystals. Subsequently, the Ti/Pt film was microstructured by photolithography (photoresist: ma-N 1420, developer: ma-D 533 S, both: Micro Resist Technology GmbH) and Ar ion beam etching, to yield a current collecting thin film grid. In the next step, ca. 250 nm thin LSF films were grown on top of the Ti/Pt grid by pulsed laser deposition (PLD) at ca. 650 °C substrate temperature. The PLD target was prepared from LSF powder (Sigma-Aldrich) by cold isostatic pressing and sintering at 1250 °C. Ablation of the LSF target was conducted by a KrF excimer laser (Compex Pro 201 F, Coherent-Lambda) with 400 mJ/pulse and 5 Hz pulse frequency.

**Experimental procedure of in-situ measurements.** In-situ XRD experiments were carried out at the Physics Hutch of beamline P07 at the PETRA 3 synchrotron (DESY, Hamburg). The diffraction experiments were carried out using an X-ray energy of 79.5 keV (equivalent to 0.01559 nm wavelength) and a beam size of 30 × 3 μm² (height × width) was used. Moreover, at this beamline, experiments using grazing incidence geometries, in combination with 2D area detectors are possible. The results here are obtained with an incident angle of 0.06°. This is close to the critical angle for total external reflection of the planar samples, which is beneficial for the signal-to-noise ratio of the diffracted intensities[49,50]. A large area Perkin Elmer integrating detector, having square 200 μm pixels, is used for data collection. Since relatively strong reflexes from the YSZ single crystal could be observed, which locally saturate the detector, beam stops were placed onto the 2D detector for protection reasons, which appear as dark areas on the 2D pattern. To avoid detector damage from the direct beam or artefacts from beryllium window Bragg reflections of the chamber, tungsten beam stops and masks were placed directly after the catalysis chamber in the exiting beam path, thereby blocking these unwanted beams.

To perform XRD measurements on the LSF thin film electrodes under electrochemical polarization at elevated temperatures and in a controlled atmosphere a commercially available catalysis diffraction chamber was used (Leiden Probe Microscopy), which was specifically designed for simultaneous solid-state electrochemistry and XRD experiments[51]. The samples were mounted inside the chamber with the CE-side facing the top plate of the sample holder (made of Inconel). For better mechanical, thermal, and electrical contact, a droplet of Pt paste was put between CE and Inconel plate, which was dried/hardened by a small torch before closing the chamber. (Please note that this way of mounting was unavoidable from a mechanical point of view, but unfortunately led to a poor gas accessibility at the CE causing a significant CE resistance in the electrochemical experiments—see Fig. 3 and the corresponding discussion.) The WE on the top side of the samples was contacted by a Pt/Ir tip pressed onto the LSF film by a piezo manipulator that is placed inside the chamber. For electrochemical polarization experiments and impedance spectroscopy measurements, electric contact was established to the WE contacting tip as well as to the CE contacting Inconel plate by means of BNC feedthroughs.

Before exsolution experiments, the samples were heated to ca. 625 °C in 5% $O_2$/balance Ar to remove potentially existing adventitious carbon from the WE surface

and to burn off residuals of organic components of the Pt paste at the CE. Under these conditions the first diffraction patterns were recorded (see Fig. 2a). After this treatment, the atmosphere was changed to 2.5% $H_2$/0.25% $H_2O$/balance Ar (total pressure 1 bar) and electrochemical as well as XRD measurements were performed.

Impedance spectroscopy and electrochemical polarization experiments were done by means of an Alpha-A High Performance Frequency Analyser with electrochemical test station POT/GAL 30V/2A (both: Novocontrol). Impedance spectra were recorded between 1 and 10 mHz with a resolution of five points per decade and with an AC voltage of 10 mV root mean square. Typical impedance spectra are shown in Supplementary Fig. 3. Since the high frequency intercept can be related to the transport resistance of oxide ions in YSZ, which usually depends exponentially on the inverse temperature, this resistance was used to derive the temperature of the samples. This procedure was already successfully applied in previous in-situ experiments[26,52] and is described in detail in refs. [53,54]. Electrochemical polarization was performed by applying a DC bias between +500 and −900 mV to the WE. The overpotential at the WE was obtained by subtracting the voltage drops at electrolyte and CE according to Eq. (8) (see below). The resulting value $\eta_{WE}$ is plotted in Fig. 3. The XRD measurements were done for various applied polarizations after the DC current had reached a stable value, thus indicating steady-state conditions.

**Lab-based electrochemical experiments.** Since the overpotential of the DC measurements at the synchrotron is subject to a relatively high error caused by the appearance of a diffusion limited CE resistance (caused by the specific sample mounting, see above), further electrochemical experiments were performed in our lab. These ex-situ electrochemical measurements were conducted on two different types of samples. On the one hand, LSF thin film electrodes after exsolution were studied (i.e., samples which were previously studied in synchrotron experiments), and on the other hand, freshly prepared LSF thin films using the same methods as described above with exception of optimised porous CEs. For the freshly prepared samples highly porous CEs were prepared consisting of multilayers of gadolinia doped ceria (GDC) and Pt: spincoated GDC paste, brushed Pt-GDC paste, and brushed Pt paste; sintered together at 1150 °C for 3 h.

The samples were electrically contacted by clamping them between two platinum sheets in an in-house made quartz setup. Current measurements using a sourcemeter (Keithley 2611B) were performed at 625 °C in an atmosphere consisting of 2.5% $H_2$, 0.25% $H_2O$, balance Ar (total pressure 1 bar). For the pre-exsolved samples, all measurements were carried out between −0.5 and +0.5 V with bias steps of 0.025 V and acquisition times of 40 s. Electrochemical bias was applied by means of three different voltage programs: (i) a sweep from −0.5 to 0.5 V (called Linear); (ii) starting from OCV with increasing bias and alternately changing polarity (denoted as Butterfly); (iii) sweeping the anodic branch up to the maximum of +0.5 V and then using the same bias steps to reach the maximum of the cathodic branch (0.5 V) before ending at OCV (called Cycles). Sketches of these voltage programs are shown as insets of the diagrams containing the respective results in Fig. 7 and Supplementary Fig. 2.

The freshly prepared samples were used for the determination of the initial exsolution point—i.e., the lowest overpotential necessary to form metallic Fe particles. Here, a constant cathodic voltage was applied for 1 h (to trigger exsolution) before recording a complete anodic branch of the I–V-curve in high resolution (to detect the switching behavior, which is characteristic for successful $Fe^0$ exsolution)—see sketch in Supplementary Fig. 5a. The anodic branch was measured from 0 to 0.4 V in 0.005 V bias steps with 30 s per step using the Linear program. After each run, the cathodic voltage was increased by −0.025 V, each time followed by recording the anodic part of the I–V-curve. The sudden appearance of a switching behavior in the anodic branch of the I–V-curve (see arrow in Supplementary Fig. 5c) identifies an overpotential η between −200 and −218 mV to be responsible for triggering the very first exsolution event. This is in good agreement with the 180 ± 30 mV at which first exsolution was observed in the in-situ XRD experiments.

**Electrochemical analysis of the I–V-curves.** The overpotential $\eta_{WE}$ and the corresponding equivalent oxygen partial pressure in the WE-bulk $p(O_2)_{WE}$ are connected according to Nernst's equation

$$\eta_{WE} = \frac{RT}{4F}\ln\left(\frac{p(O_2)_{WE}}{p(O_2)_{atm}}\right) \tag{7}$$

with $p(O_2)_{atm}$ denoting the oxygen partial pressure in the atmosphere; R, T, and F are gas constant, absolute temperature, and Faraday's constant, respectively. The WE overpotential $\eta_{WE}$ is smaller than the voltage applied to the cell. The latter also contains an ohmic contribution from the electrolyte conductivity, and an electrochemical contribution from the CE.

For the ideal case of a kinetically very fast CE $\eta_{WE}$ can be obtained by simply subtracting the ohmic drop in the electrolyte from the applied voltage $U_{set}$—this was the case for the electrochemical ex-situ measurements. In the in-situ experiments, however, the CE also showed a significant polarization resistance (which was most likely caused by a gas diffusion limitation at the porous CE due to the necessary way of mounting the sample). Thus, the overpotential at the WE was

obtained by

$$\eta_{WE} = U_{set} - R_{YSZ} I_{DC} - \eta_{CE} \qquad (8)$$

with $R_{YSZ}$ being the resistance of ion conduction in the YSZ electrolyte obtained by impedance spectroscopy (high frequency intercept in Supplementary Fig. 3), $I_{DC}$ the DC current flowing through the cell under polarization, and $\eta_{CE}$ the voltage drop at the CE. The latter is caused by a nonlinear resistor and $\eta_{CE}$ was estimated from the bias dependent CE resistances obtained by impedance spectroscopy. This estimate causes the relatively large error bars of the current–voltage (I–$\eta_{WE}$) curve in Fig. 3.

The measured DC current upon electrochemical polarization is directly proportional to the net reaction rate of hydrogen oxidation $r_{net}$ by Faraday's law and is given by

$$I_{DC} = 2Fr_{net} = 2F(\overrightarrow{r} - \overleftarrow{r}). \qquad (9)$$

The net reaction rate is the difference of the forward and backward reaction rates $\overrightarrow{r}$ and $\overleftarrow{r}$ of the WE reaction (i.e., Eq. (1)), which are the rates of anodic hydrogen oxidation and cathodic water splitting, respectively.

To further illustrate the different electrochemical behavior of the LSF WEs in the two regimes, a fitting procedure was conducted by assuming exponential shapes for both partial curves. This means, that for the sake of simplicity, we neglected the derivation from an exponential curve that becomes visible in the lab-based, more detailed I–V-measurements especially at higher cathodic overpotentials. For fitting the high activity part of the I–V-curve (between ca. −400 and +150 mV)

$$I = I_0 \left( e^{\beta \frac{F}{RT} \eta} - e^{-\beta \frac{F}{RT} \eta} \right) \qquad (10)$$

was used. Therein, the fitting parameter $I_0$ is the exchange current density of the observed reaction and thus directly reflects the catalytic activity of the LSF surface for $H_2$ oxidation/$H_2O$ splitting. The parameter $\beta$ depends on the reaction mechanism[28,55] and was fixed to 0.5. This corresponds to a symmetric fit function (i.e., hyperbolic sine) and was done to yield a stable fit, since otherwise the relatively few data points in the anodic part would be underrepresented by the fit result. For the low activity part of the I–V-curve (between ca. +150 and +400 mV) only the anodic branch of Eq. (10) was considered—i.e.,

$$I = I_0' \cdot e^{\beta' \frac{F}{RT} \eta} \qquad (11)$$

was used for fitting. Here, the parameter $\beta'$ had to be set free to yield a meaningful fit. The resulting fit parameters obtained for the two partial curves are summarized in Supplementary Table 1. The ratio of the two obtained exchange current densities almost exactly matches the ratio between the two polarization resistances of the WE measured by impedance spectroscopy (without bias) before and after iron exsolution (cf. Supplementary Fig. 3). This strongly supports the consistency of the electrochemical results. The significantly different $\beta$-value for the high and low activity part of the curve is in line with the interpretation that the step in the current–voltage curve is indeed associated with a change of the reaction mechanism at the LSF surface due to the presence of metallic iron particles.

**Phase identification**. XRD patterns were analysed by comparing the experimentally obtained diffractograms with powder patterns from literature[31]. However, this database comparison alone was not sufficient for a distinct phase assignment, since several Pt and LSF reflexes overlap in their 2Θ diffraction angle. Separation was though possible in the 2D pattern due to the different texture of both films, which lead to a segmented appearance of the observed Debye–Scherrer rings for the different phases. Moreover, phase identification was supported by a comparison of the diffraction patterns measured in oxidizing atmosphere and immediately after switching to reducing conditions (Fig. 2a, b, respectively). In reducing atmosphere, oxygen is released from the LSF lattice and cations become reduced, which causes a significant lattice expansion[56,57]. This effect is commonly called chemical expansion and is here very helpful to distinguish the perovskite-type LSF pattern from the metallic Pt, which does not show such a type of lattice parameter change upon $p(O_2)$ variation. On LSF an increase of the out-of-plane lattice constant of 0.69% was found. For a more detailed phase analysis of standard 1D diffractograms, generated from the 2D diffraction patterns by integration over the angle χ, the reader is referred to Supplementary Fig. 1.

**Analysis of $Fe^0$ particle growth**. The α-Fe (110) peak profile was analysed by fitting a Gaussian to it and calculating the average particle size by using Scherrer's formula[32]

$$d = \frac{K \cdot \lambda}{\sigma \cdot \cos(\Theta)} \qquad (12)$$

with $d$ being the particle size, $K$ a constant factor, $\lambda$ the wavelength of the X-ray radiation, $\sigma$ the full width at half maximum of the peak, and $\Theta$ one half of the diffraction angle. The value for $K$ depends on the geometrical particle shape and varies from 0.88 to 1.1. Here we use an average value $K = 1$, since the exact shape is not known and we are mainly interested in the trend of the size evolution rather than the exact size of the particles.

In literature, three factors have been discussed as possibly determining the growth process of metallic exsolution particles. The particle growth rate can be limited by the rate of iron reduction at the surface. For such reduction-controlled kinetics two limiting cases can be distinguished, growth either limited (i) by the concentration of the reactant or (ii) by strain caused due to the strong interaction of metal and oxide, which induces a size dependence of the activation energy of iron reduction. (iii) In case of a fast reaction but a finite supply of exsolvable cations from the bulk of the perovskite to the surface, a diffusion limited growth is found. The corresponding fit equations representing these three models were taken from a paper by Gao et al[33]. and the corresponding fit functions assuming spherical particles are given by Eqs. (13)–(15) below:

- Strain-controlled kinetics

$$d = d_{0,s} \left( \ln \left( 1 + \frac{t}{\tau_s} \right) \right)^{\frac{1}{3}}, \qquad (13)$$

- Reactant-controlled kinetics

$$d = d_{0,r} \left( 1 - e^{\left( -\frac{t}{\tau_r} \right)} \right)^{\frac{1}{3}}, \qquad (14)$$

- Diffusion-controlled kinetics

$$d = d_{0,d} \left( \frac{t}{\tau_d} \right)^{\frac{1}{6}}. \qquad (15)$$

In these equations $d$ and $t$ denote particle diameter (assuming spherical particles) and time from first exsolution, respectively; $d_0$ and $\tau$ are fitting parameters.

The fit result is depicted in Supplementary Fig. 6 and suggests that the growth rate of the particles is limited by the rate of iron reduction at the surface of LSF. This result is also in line with studies in literature, which also found a reaction-controlled growth of metallic particles exsolved from perovskite-type oxides[21,34]. However, for the exsolution of $Fe^0$ from LSF a distinction between strain- and reactant-controlled is not possible, but since Fe is the only element on the B-site of LSF and its high concentration does not change drastically upon exsolution, a minor (if any) role of the reactant concentration for the particle growth rate can be assumed. Hence, for the system here, the growth of the $Fe^0$ particles appears to be significantly affected by the strain of the particles.

**Electron microscopy**. To maintain the metallic state of the particles for the SEM measurements as good as possible, the respective sample was cooled in $H_2$:$H_2O$ = 10:1 atmosphere to room temperature after the last electrochemical measurement and immediately transferred under Ar atmosphere to the SEM and focused ion beam (FIB) machine to prepare the TEM lamella. SEM was performed in secondary electron yield mode on a FEI Quanta 250 FEG. Images were recorded close to the embedded Pt current collectors of the LSF WEs to avoid charging of the samples, since coating with a conductive film was omitted to avoid artefacts in the subsequent TEM analysis.

Preparation of the TEM lamella was done by FIB cutting on an FEI Quanta 200 3D dual-beam system. Following a typical in-situ lift-out FIB TEM sample preparation recipe[58], a Pt protection layer of about 150 nm thickness was deposited by electron beam induced deposition to protect the near surface region of the sample. The thickness of the protection layer was increased to about 3 µm by an ion beam induced deposition of Pt. The preparation proceeds by removing bulk material around the protected area with the Ga ion beam, ending up in an about 2 µm thick lamella. The lamella was freed from bulk by undercutting, followed by lifting the lamella out, and welding it to a TEM sample grid. In consecutive milling steps with decreasing ion currents—we have used 1 nA and 500 pA for the medium thinning and 100 pA for fine milling—the thickness of the lamella was reduced to about 120 nm. In order to decrease the FIB induced damage, generated by the previous milling steps at 30 kV ion acceleration voltage, a final cleaning of the lamella was done with 5 kV and 30 pA.

TEM imaging was performed on a FEI TECNAI F20 field emission TEM, operated at 200 kV. Bright field TEM images were recorded using a Gatan RIO 16 CMOS camera. STEM images were obtained with a high-angle annular dark field (HAADF) detector. Energy dispersive X-ray spectroscopy (EDX) elemental maps were acquired with a windowless EDAX Apollo XLTW silicon drift detector collecting intensities of the La–L, Sr–L, Fe–K, and O–K peaks. Supplementary Fig. 8 shows the result of elemental mapping together with an overview image acquired with an HAADF in scanning transmission mode (STEM). The HAADF micrograph in Supplementary Fig. 8a shows negligible diffraction contrast. This makes an interesting property of the post-exsolved films visible: They contain numerous closed pores with sizes in the nanometer range. These pores are visible as black dots within the LSF film and occur particularly frequently at the interface. Since such a nano-porosity was never observed so far for our PLD-grown perovskite-type thin films[59–61], it is very likely connected with the exsolution of iron. In view of the relatively large amount of exsolved iron a connection of the pore formation with the exsolution process is reasonable—maybe by a Kirkendall-like mechanism. Indeed Kirkendall-porosity formation has been already reported

for perovskite-type diffusion couples[62,63], but a detailed clarification of the pore formation mechanism upon exsolution needs to be done in future studies.

The distribution of Fe, La, Sr, and O were recorded for the two regions marked by the dashed and solid box in Supplementary Fig. 8a. For the dashed box the results are depicted in Supplementary Fig. 8b–e. It confirms that each of the surface particles is significantly enriched in iron. For the smaller solid box, the distribution of Fe, La, Sr, and O is shown in Supplementary Fig. 8f–i. Comparison of these elemental maps shows a clear depletion of the La, Sr, and O content for the Fe-enriched region of the particle. This picture obtained by EDX elemental mapping is consistent with the interpretation of XRD data that exsolution yields only iron particles at the LSF surface and neither Sr nor La enriched particles were found.

## Data availability

Source data file available as supplementary material.

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

## Acknowledgements

The authors gratefully acknowledge funding by the Austrian Science Fund (FWF) through project F4509-N16 as well as DESY (Hamburg, Germany), a member of the Helmholtz Association HGF, for the allocation of beamtime and provision of experimental facilities.

## Author contributions

A.K.O. performed the experiments at the beamline, analysed the electrochemical as well as XRD. data, and wrote the manuscript. A.N. wrote the beamtime-proposal, performed the experiments at the beamline, and analysed the XRD data. V.V. performed the experiments at the beamline and contributed in analysing the XRD data. S.V. prepared the experimental setup and performed the experiments at the beamline. F.B. prepared the setup at the beamline and supervised the beamtime as beamline scientist. H.S. performed the lab-based electrochemical experiments. S.S. recorded TEM images and elemental mappings by EDX. A.S.T. prepared the FIB-lamella and recorded the SEM images. J.B. recorded TEM images and supervised analysis of electron microscopy data. A.S. performed and supervised the experiments at the beamline. J.F. acquired the funding, supervised the data analysis, and contributed to writing the manuscript.

## Competing interests

The authors declare no competing interests.
