## [Peer Review File · Nature Communications]

Reviewers' comments:

Reviewer #1 (Remarks to the Author):

The paper is in general well addressed and present very relevant information regarding a very important topic such as exsolution of metallic particles during operation under cathodic and anodic polarization. The findings are of interest, however in my opinion the work should be complemented with HR-TEM in order to confirm the formation of the Fe particles. In addition, here are some additional comments to be corrected/clarified before publication:

- 1) Page 5 line 82: What is the reason for having a gradient in temperature of 20 °C? this is linked with my question about EIS data.
- 2) Figure 3, line 302 (and the explanation in the text). Although the "jump effect" observed when Fe-oxides are formed at voltages of about 150 mV, this effect is not clearly explained. From the experimental section, it is not easy to understand how the experiment was conducted. Steady state experiments at a fixed voltage for how long? What would have happen if longer stabilization periods were used? Could Fe-oxides be formed at lower voltages? Is there any different by running first in SOFC and the SOEC or viceversa? Of after chronoamperometry combining SOFC and SOEC modes?
- 3) Page 19, lines 388-390: this statement is also explained in the supplementary information. I suggest to remove this from the latter.
- 4) Line 392. DC bias up to -900 mV were applied, however the authors only show data up to about -350 mV, could you please comment about this data?
- 5) Regarding the ohmic contribution of the two samples in figure S3, how the authors are sure that this difference is only due to different temperatures? Why this temperature can be different? Could be also related to electrical contacts or lateral current collection?
- 6) Also in figure S3, what about the data at lower frequencies for the "metallic Fe sample"? Is this points also due to R (CE) as for the pristine sample? If that is the case, why is so different? I think you should have measured to lower frequencies, as 10 mHz is not such a low frequency.
- 7) After figure S3, authors comment that they have measured EIS under cathodic polarization but the data is not shown. EIS spectra should be shown here.

Reviewer #2 (Remarks to the Author):

With the help of synchrotron-based operando X-Ray diffraction, this study has successfully clarified how catalytic Fe nanoparticles exsolved from LSF perovskites can be switched reversibly between high and low active states at changing applied voltage. It also reveals that even after re-oxidation, the exsolved Fe particles become Fe-oxides instead of dissolved into parent perovskite. The mechanism of how water electrolysis activity is improved by exsolution is also discussed deeply. However, the data in the manuscript is not sufficient enough to support authors' conclusions. Either more experimental or simulation work has to be conducted to complete the work. Therefore, considering Nature Communication's requirements on high quality and novelty, rejection is recommended. Authors may submit this manuscript to some other journals focusing on electrochemistry. The authors may consider the following suggestions for revision.

1. More evidence has to be provided to confirm the growth of iron nanoparticles is a diffusion-controlled process. Only a $r(\text{size}) \sim t^{0.5}$ curve without root mean square error shown is not sufficient to prove it. I would suggest the authors to apply a diffusion-controlled coarsening model to estimate the exact value of diffusion coefficient and compare it with the value reported in the literature. If they are comparable, then it is safe to conclude that the growth is a diffusion-controlled process.
2. How did the authors determine that 180 mV is the initial voltage for exsolution of iron nanoparticles?

3. Is it possible to add images showing the microstructure evolution (exsolution, dissolution of iron particles) while switching between low and high activities? This could make the conclusion more convincing and the manuscript more valuable.
4. Results are needed to support the conclusion "the size of particles plays a minor role for the activity enhancement", for example, the activity (current) and time (particle size) relationship.
5. On Line 40, Page 3, it not accurate to say "The first step of such energy storage processes is water electrolysis" since the first step could also be electrolysis of carbon dioxide and others.
6. When discussing the mechanism and interpretation of the catalytic effect of exsolved Fe nanoparticles, authors stated that existence of iron nanoparticles enhanced the formation of hydroxyl,

which is the rate-limiting step of water electrolysis. As is shown below, dissociative adsorption of H₂ on Fe is much faster than that on LSF and absorbed H atom (Had,Fe) is very likely to exchange with absorbed H atom (Had,LSF)

However, this conclusion is completely based on the results reported in the literature. No experimental results from this work is provided. This is not sufficient to include this as a conclusion in this manuscript. Even if the experimental work is difficult to be done, at least simulation or calculation should be provided. For example, the dissociative adsorption energy of H₂ on Fe and LSF can be determined by DFT calculations and compared. Also is it possible to measure or calculate the reaction rate constant of each step of water electrolysis with exsolved iron particles to confirm the rate limiting step?

Response to the comments of the reviewers and modifications made

Manuscript number: NCOMMS-19-29493-T

How do exsolution catalysts become electrochemically switchable?

Reviewer #1:

The paper is in general well addressed and present very relevant information regarding a very important topic such as exsolution of metallic particles during operation under cathodic and anodic polarization. The findings are of interest, however in my opinion the work should be complemented with HR-TEM in order to confirm the formation of the Fe particles.

Authors:

We thank the reviewer for the positive feedback and we agree that TEM measurements are a very helpful complementary method to substantiate our statements. We thus recorded SEM and TEM images on our thin films after exsolution, which we include in the paper in an additional section. Moreover, the “Methods” section and the supporting information were adapted/extended – changes/additions are highlighted by text in blue color.

The list of authors was extended accordingly by the following persons: Andreas Steiger-Thirsfeld (recording of SEM images, FIB-cutting and preparation of TEM lamella), Sabine Schwarz (recording of TEM images), Johannes Bernardi (recording of TEM images & interpretation of TEM data).

Reviewer #1:

In addition, here are some additional comments to be corrected/clarified before publication:

Page 5 line 82: What is the reason for having a gradient in temperature of 20 °C? This is linked with my question about EIS data.

Regarding the ohmic contribution of the two samples in figure S3, how the authors are sure that this difference is only due to different temperatures? Why this temperature can be different? Could be also related to electrical contacts or lateral current collection?

Authors:

The ± 10 °C of the experimental temperature do not refer to a gradient in temperature within the sample, but to the possible accuracy of temperature measurement. This error arises from several contributions: (i) The temperature of the heater used in the setup at the beamline showed slight drifts during the course of the measurement. Moreover, as the reviewer correctly states, the ohmic resistance of the impedance spectrum can also arise from (ii) different contact resistance for different samples or (iii) different current collection resistance even for one sample but different polarisation or different atmosphere, since both change the defect chemistry of the LSF film and thus its electronic conductivity. To be able to

state one experimental temperature for all samples, temperatures, atmospheres and electrochemical polarisations we decided to give the average temperature with a moderate error, i.e. 625 ± 10 °C.

Regarding the slightly different ohmic resistance of the two spectra in Fig. S3: Since both spectra were measured on the same sample, we can be rather sure to have the same contact and current collection resistance for both spectra. This was expressed misleadingly in the SI and we thus revised this sentence. However, the reviewer may be correct for the case that exsolution causes a noteworthy change of the electronic conductivity of the WE film. To consider this fact, we adapted the corresponding statement in the SI accordingly (changes made are highlighted by blue text).

Reviewer #1:

Figure 3, line 302 (and the explanation in the text). Although the “jump effect” observed when Fe-oxides are formed at voltages of about 150 mV, this effect is not clearly explained. From the experimental section, it is not easy to understand how the experiment was conducted. Steady state experiments at a fixed voltage for how long? What would have happen if longer stabilization periods were used? Could Fe-oxides be formed at lower voltages? Is there any difference by running first in SOFC and the SOEC or viceversa? Or after chronoamperometry combining SOFC and SOEC modes?

Authors:

Each of the points in Fig. 3 represents an average of the steady state DC current measured upon applying a certain DC voltage. For each point, the current was averaged over at least 20 seconds, in most cases the steady state duration was even longer, depending on the XRD experiments (and possible optimizations of the diffractometer necessary to improve the quality of diffraction data). Owing to the relative long duration of the entire experiment and the related possibility of degradation of the WE, we did not measure the I-V curve by simply increasing the set-voltage step by step. Rather, we defined a random measurement program, which included changing signs of polarity to exclude time dependencies of the curve. We changed the corresponding text in the manuscript accordingly (changes made are highlighted by blue text).

The further questions of the reviewer regarding effects of time and pre-history of the samples are indeed very interesting, since the behaviour of the samples at the synchrotron suggests kinetic effects to play a role for the exsolution process. They triggered us to study the “old” synchrotron samples together with freshly prepared ones in our lab. Accordingly, Harald Summerer, who performed these electrochemical experiments, was added to the list of authors.

Moreover, we have to admit that these questions of the reviewer and our related data analysis were very helpful in finding an error in Fig. 3 – the current was related to the wrong area, which is corrected now.

Reviewer #1:

Page 19, lines 388-390: this statement is also explained in the supplementary information. I suggest to remove this from the latter.

Authors:

We agree with the reviewer and removed this part from the SI.

Reviewer #1:

Line 392. DC bias up to -900 mV were applied, however the authors only show data up to about -350 mV, could you please comment about this data?

Authors:

The -900 mV bias are the set voltage (U_{set} in Eq. S2), while the -350 mV are the overpotential η_{WE} , which drops at the working electrode. We wanted to express this fact by the sentence *“The overpotential dropping at the WE was obtained by subtracting the voltage drops at electrolyte and counter electrode according to Eq. S2 (see supporting info).”*

To make the statement clearer we modify it as follows: *“The overpotential at the WE was obtained by subtracting the voltage drops at electrolyte and counter electrode from the set voltage U_{set} according to Eq. S2 (see supporting info). The resulting value η_{WE} is plotted in Fig 3.”*

Reviewer #1:

Also in figure S3, what about the data at lower frequencies for the “metallic Fe sample”? Is this points also due to R (CE) as for the pristine sample? If that is the case, why is so different? I think you should have measured to lower frequencies, as 10 mHz is not such a low frequency.

Authors:

In principle we agree with the referee that impedance measurements to even lower frequencies – e.g. 1 mHz – would be desirable. However, one has to bear in mind that in this particular case one only gains 5 additional frequency points in the spectrum for the cost of a 10-fold measurement time. Moreover, one needs to consider that only one sine wave at 1 mHz has a duration of 16.7 min, hence recording of impedance spectra would take extremely long. Since time is very precious in synchrotron measurements, the WE feature is anyway well separated, and the information gained from the CE feature does not play a big role for the interpretation of the effects at the WE, we decided not to measure at lower frequencies. Please also note, that this decision had to be made during the beamtime, since the CE feature was a direct consequence of mounting the sample on the heater of the beamline and did never appear in lab experiments. One disadvantage of this is the resulting error of the overpotential (see error bars in Fig. 3).

The somewhat different shape of the counter electrode feature for both spectra in Fig. S3 may be attributed to Fe exsolution, since the porous CE was also based on LSF. However, owing to the limited number of frequency points, this is rather speculations. Since this question is nevertheless irrelevant, we want to avoid potentially confusing discussion on this topic in the paper.

Reviewer #1:

After figure S3, authors comment that they have measured EIS under cathodic polarization but the data is not shown. EIS spectra should be shown here.

Authors:

We did not intend to show impedance data under polarisation, since interpretation is rather challenging and effects of exsolution and polarisation on the electrode impedance can hardly be separated. It seems that the corresponding sentence was misinterpreted by the reviewer and we thus changed it to avoid ambiguity (changes made are highlighted by blue text).

Reviewer #2:

With the help of synchrotron-based operando X-Ray diffraction, this study has successfully clarified how catalytic Fe nanoparticles exsolved from LSF perovskites can be switched reversibly between high and low active states at changing applied voltage. It also reveals that even after re-oxidation, the exsolved Fe particles become Fe-oxides instead of dissolved into parent perovskite. The mechanism of how water electrolysis activity is improved by exsolution is also discussed deeply. However, the data in the manuscript is not sufficient enough to support authors' conclusions. Either more experimental or simulation work has to be conducted to complete the work.

Authors:

We recorded SEM and TEM images and conducted additional lab-based electrochemical experiments to provide further experimental evidence supporting our conclusions. Based on these data we have modified the manuscript substantially, hoping that the reviewer now finds our results convincing (see also our response to reviewer #1).

Reviewer #2:

More evidence has to be provided to confirm the growth of iron nanoparticles is a diffusion-controlled process. Only a $r(\text{size}) \sim t^{0.5}$ curve without root mean square error shown is not sufficient to prove it. I would suggest the authors to apply a diffusion-controlled coarsening model to estimate the exact value of diffusion coefficient and compare it with the value reported in the literature. If they are comparable, then it is safe to conclude that the growth is a diffusion-controlled process.

Authors:

We agree with the reviewer, that analysis by a $t^{0.5}$ curve is not sufficient to give an indication of the rate limitation of particle growth. We therefore analysed the XRD data together with the particle size obtained by SEM (done after additional lab-based electrochemical experiments) using different growth models described in literature [Y. Gao et al. Nano Energy 27 (2016), 499].

The respective figure and the related text were changed according to the results of this analysis – changes made to the manuscript are given in blue.

Reviewer #2:

How did the authors determine that 180 mV (*sic*) is the initial voltage for exsolution of iron nanoparticles?

Authors:

The -180 mV was the first cathodic polarisation we applied after investigating the LSF film for 13 min in H₂/H₂O atmosphere without polarisation. That's what we wanted to express by stating "...*the very first exsolution of iron from the virgin LSF film only occurred after applying a sufficiently high voltage,...*". The applied voltage was chosen based on our previous experience with these electrodes, with the intention to apply a sufficiently large cathodic voltage to trigger exsolution. To give more detailed insights in the initial exsolution process, we conducted further lab-based electrochemical experiments, which are included in the revised version of the supporting information (see also our answer to reviewer

#1 above). As one can see from these data, the initial polarisation of $\eta_{WE} = -180$ mV during the beamtime is rather close to the point of initial exsolution in lab-based experiments.

Reviewer #2:

Is it possible to add images showing the microstructure evolution (exsolution, dissolution of iron particles) while switching between low and high activities? This could make the conclusion more convincing and the manuscript more valuable.

Authors:

Corresponding XRD images recorded *in-situ* while switching between Fe^0 and Fe-oxides are shown in Fig. S2 as well as Figs. 4 & 5, which were derived from the raw data in Fig S2 by integration. To make this fact clearer we changed the numbers (1-6) indicating chronology to the distinct time at which the respective data was recorded.

In-situ recorded optical images of the switching process, would require electrochemical experiments in the electron microscope, which is beyond the scope of the present work. Successful realisation would certainly justify at least one separate paper. Instead, we performed *ex-situ* TEM investigations to visualise the exsolved particles and included the results in our manuscript (see also our response to reviewer #1 above).

Reviewer #2:

Results are needed to support the conclusion “the size of particles plays a minor role for the activity enhancement”, for example, the activity (current) and time (particle size) relationship.

Authors:

In this point we respectfully disagree with the reviewer, since the current and particle size relationship is already provided. The I-V-curve was measured over a period of ca. 3.5 hours during which the particles grew from ca. 5 to almost 20 nm. Current density changes caused by this growth are within the experimental error of the I-V-curve and significantly smaller than the factor of 4 of the particle size. Hence, we concluded a “minor role” of the particle size. Effects within the experimental error of our data are of course possible.

The change of chronology notation mentioned above, helps making this relationship clearer. Moreover, we adapted the corresponding text to make it easier understandable.

Reviewer #2:

On Line 40, Page 3, it not accurate to say “The first step of such energy storage processes is water electrolysis” since the first step could also be electrolysis of carbon dioxide and others.

Authors:

We agree with the reviewer, that this was inaccurate and changed the sentence to: “*The first step of such energy storage processes is most likely electrolysis of water or carbon dioxide, which can be done in solid oxide cells with highest thermodynamic efficiency.*”

Reviewer #2:

When discussing the mechanism and interpretation of the catalytic effect of exsolved Fe nanoparticles, authors stated that existence of iron nanoparticles enhanced the formation of hydroxyl,

which is the rate-limiting step of water electrolysis. As is shown below, dissociative adsorption of H₂ on Fe is much faster than that on LSF and absorbed H atom (H_{ad,Fe}) is very likely to exchange with absorbed H atom (H_{ad,LSF})

However, this conclusion is completely based on the results reported in the literature. No experimental results from this work is provided. This is not sufficient to include this as a conclusion in this manuscript. Even if the experimental work is difficult to be done, at least simulation or calculation should be provided. For example, the dissociative adsorption energy of H₂ on Fe and LSF can be determined by DFT calculations and compared. Also is it possible to measure or calculate the reaction rate constant of each step of water electrolysis with exsolved iron particles to confirm the rate limiting step?

Authors:

This comment is most probably due to a misunderstanding of what we intended to say. It was not our intention identifying the rate-determining step of H₂ oxidation on the perovskite oxide, but to explain how the electrochemical switching of surface particles changes the catalytic activity of the LSF electrode.

However, the comment helped us identifying two weak points of our manuscript, which may have triggered the reviewer’s criticism; both points were modified in the revised manuscript. First, the respective paragraph in the conclusion was adapted to unambiguously communicate the mechanistic message formulated above. Second, more references are now included in the discussion of the mechanism change upon switching, to acknowledge literature data accordingly. Since the data situation for hydrogen surface species on Fe and LaFeO₃-based perovskites is already good, performing additional DFT simulations would not lead to much novel results and we decided to support our discussion by citation of already existing DFT studies with high quality.

REVIEWERS' COMMENTS:

Reviewer #1 (Remarks to the Author):

Authors have replied to all the concerns and the paper is recommended for publication.

Reviewer #2 (Remarks to the Author):

For my Q3, I understand that in-situ record of the images of the switching process is beyond the scope of the authors' present work.

For my Q6, I was asking authors to provide evidence to confirm the existence of the spillover on exsolved Fe nanoparticles and exchange of *H between Fe nanoparticles and LSF. Although the authors has not provided the direct evidence from experimental work or theoretical calculation, they provided several references to support their statement.

Therefore, I think authors have already addressed all the concerns raised by the reviewers and I recommend acceptance of the manuscript in the current form.

Jing-Li Luo